

# Collaborative clinical reasoning: a scoping review

Ching-Yi Lee[1,*], Hung-Yi Lai[1,*], Ching-Hsin Lee[2], Mi-Mi Chen[1] and Sze-Yuen Yau[3]

[1] Department of Neurosurgery, Chang Gung Memorial Hospital at Linkou and Chang Gung University College of Medicine, Taoyuan, Taiwan
[2] Department of Radiation Oncology, Proton and Radiation Therapy Center, Chang Gung Memorial Hospital at Linkou, Taoyuan, Taiwan
[3] (CG-MERC) Chang Gung Medical Education Research Centre, Linkou, Taoyuan, Taiwan
[*] These authors contributed equally to this work.

Corresponding author
Ching-Yi Lee, 8702021@cgmh.org.tw

## ABSTRACT

**Background.** Collaborative clinical reasoning (CCR) among healthcare professionals is crucial for maximizing clinical outcomes and patient safety. This scoping review explores CCR to address the gap in understanding its definition, structure, and implications.

**Methods.** A scoping review was undertaken to examine CCR related studies in healthcare. Medline, PsychInfo, SciVerse Scopus, and Web of Science were searched. Inclusion criteria included full-text articles published between 2011 to 2020. Search terms included cooperative, collaborative, shared, team, collective, reasoning, problem solving, decision making, combined with clinical or medicine or medical, but excluded shared decision making.

**Results.** A total of 24 articles were identified in the review. The review reveals a growing interest in CCR, with 14 articles emphasizing the decision-making process, five using Multidisciplinary Team-Metric for the Observation of Decision Making (MDTs-MODe), three exploring CCR theory, and two focusing on the problem-solving process. Communication, trust, and team dynamics emerge as key influencers in healthcare decision-making. Notably, only two articles provide specific CCR definitions.

**Conclusions.** While decision-making processes dominate CCR studies, a notable gap exists in defining and structuring CCR. Explicit theoretical frameworks, such as those proposed by Blondon et al. and Kiesewetter et al., are crucial for advancing research and understanding CCR dynamics within collaborative teams. This scoping review provides a comprehensive overview of CCR research, revealing a growing interest and diversity in the field. The review emphasizes the need for explicit theoretical frameworks, citing Blondon et al. and Kiesewetter et al. The broader landscape of interprofessional collaboration and clinical reasoning requires exploration.

## BACKGROUND

### Clinical reasoning errors

Diagnostic errors pose a significant challenge in healthcare, with an estimated error rate of 10% to 15% according to autopsy data in the United States (*Graber, 2013*; *Shojania et al., 2003*). While diagnostic errors are not the primary cause of death in the country, they still exert a substantial impact on patient outcomes and healthcare costs. The majority of errors that occur can be attributed, at least in part, to cognitive processes of individual healthcare professionals (*Norman & Eva, 2010*). Faulty clinical reasoning is considered a key contributor to diagnostic errors, and studies suggest that error prevention requires an improvement in clinical reasoning skills (*Connor, Durning & Rencic, 2020*; *Durning, Trowbridge & Schuwirth, 2020*; *Norman et al., 2017*).

### Clinical reasoning in medical education

Clinical reasoning, a central component of professional competence for healthcare practitioners, is defined as "the thought process that guides practice" (*Connor, Durning & Rencic, 2020*). Terms such as problem-solving, decision-making, critical thinking, and judgment are also used interchangeably with clinical reasoning (*Norman, 2005*). This process involves collecting cues, processing information, understanding patient problems or situations, planning and implementing interventions, evaluating outcomes, and reflecting on and learning from the entire process (*Levett-Jones et al., 2010*).

One influential model explaining clinical reasoning process is the "dual-process" theory of cognition, which posits that errors are often associated with "system 1" thinking (automatic and intuitive) rooted in cognitive heuristics (*Royce, Hayes & Schwartzstein, 2019*). Although "system 1" thinking allow for rapid judgement through pattern recognition, it is susceptible to the biases and emotional influences (*Schwartz & Elstein, 2008*). On the other hand, "system 2" thinking (slow, effortful, and analytic) can yield more normatively rational reasoning, but it is easily disrupted by high cognitive loads (*Evans, 2008*; *Norman & Eva, 2010*). In busy clinical settings, where continuous system 2 thinking is impractical, healthcare practitioners often rely on system 1 thinking, which may lead to incomplete or incorrect diagnoses and practices.

### Interprofessional collaborative clinical reasoning

While previous literature on collaborative healthcare has primarily focused on teamwork competencies and interprofessional collaboration (*Figueroa et al., 2013*; *Ponte et al., 2010*; *Shrader et al., 2013*), the current study seeks to explore collaborative clinical reasoning. The conventional discussion of team impacts on healthcare professional competences mainly focused on individualist discourse. They emphasized on the outcomes, with the individual gain that practitioners acquire, perform, and maintain over their practice life. The notion of "collective competencies" shed light on the underlying mechanism of teamwork (*Anderson, 2012*). It addresses how individually "incompetent" healthcare professionals shared and distributed to form a "competent" team. This collectivist discourse focuses on the similarities and differences that each practitioner perceived in the situation, and how they trigger and share the mental models among the various team members. The term

"collaborative reasoning" proposed by Mason will be employed to describe the process of reaching a shared mental model (*Mason & Santi, 1998*). It was proposed that team participants work together efficiently by anticipating other members' responses. One of the insights was that the degree to which team members shared to develop a shared mental model is positively correlated with the team performance (*Lim & Klein, 2006*).

## Significance of current study

The dual-processing model of clinical reasoning involves both systems 1 and 2 thinking during decision-making among healthcare professionals. While system 1 thinking is advantageous for quick judgments, system 2 thinking is less effort-prone but demands more mental effort. In a busy clinical setting, it is impractical for an individual healthcare professional to stay in system 2 thinking continuously, despite this type of thinking is often crucial and less prone to error (*Baddeley, 1992*; *Evans, 2008*; *Schwartz & Elstein, 2008*). Collaborative clinical reasoning, akin to shared mental models, may facilitate cognitive load sharing in a complex situation involving multiple healthcare professionals. It may help identify, reduce subjective biases and leads to efficient decision-making during diagnostic processes through team effort and communication (*Anderson, 2012*; *Figueroa et al., 2013*; *Lim & Klein, 2006*; *Mason & Santi, 1998*). A preliminary scholarly search has indicated a scarcity of literature on collaborative performance in clinical reasoning and most studies only address the importance of communication in a healthcare team or describe team effort with the common goal of reaching a consensus for decision making (*Anderson, 2012*; *Figueroa et al., 2013*; *Kiesewetter, Fischer & Fischer, 2017*; *Lim & Klein, 2006*; *Mason & Santi, 1998*). Amidst this scholarly landscape, the term "multidisciplinary teams or meetings" (MDTs or MDMs) emerges as a recurring theme within the literature addressing collaboration in healthcare. MDTs or MDMs are structured gatherings involving professionals from various disciplines within the healthcare setting. These meetings serve as a platform for collaborative decision-making and comprehensive assessment of complex cases involving patients. However, a notable trend surfaces—much of the literature leans heavily towards quantitative assessments. The focus on MDTs or MDMs tends to revolve around numerical evaluations, leaving a gap in our comprehension of how collaborative clinical reasoning shapes both system 1 and system 2 thinking. A further insight into the cognitive process or the diagnostic dimension of collaborative clinical reasoning is therefore required. This review, therefore, aims to address this scholarly gap by systematically mapping the available evidence. Our goal is to provide a thorough understanding of how multidisciplinary healthcare professionals engage in collaborative clinical reasoning, shedding light on its cognitive underpinnings and its implications for informed decision-making. As we map CCR research, we aim to answer the following research questions:

RQ1: What is the current status of collaborative clinical reasoning (CCR) research in general?

RQ2: How is collaborative clinical reasoning conceptualized and practiced within multidisciplinary teams or meetings (MDTs or MDMs)?

# METHOD

In accordance with the Arksey and O'Malley framework (*Arksey & O'Malley, 2005*), and the recent recommendations by *Levac, Colquhoun & O'Brien (2010)*, our scoping review methodology comprises the following steps: (1) scoping review questions, (2) search strategy, (3) study screening and selection, (4) data extraction, (5) analysis and presentation of results, and (6) team consultation.

1. Review questions
   This review is centered around two overarching question, "What is the current status of collaborative clinical reasoning (CCR) research in general?" and "How is collaborative clinical reasoning conceptualized and practiced within multidisciplinary teams or meetings (MDTs or MDMs)?"

2. Relevant studies and search strategy
   The search involved four electronic databases: Medline, PsychInfo, SciVerse Scopus (multidisciplinary, 1823-present), and Web of Science (multidisciplinary, 1900-present). We limited the search to the years between 2011 and 2020. The language of articles is limited to English. Using Kiesewetter's search strategy (*Kiesewetter, Fischer & Fischer, 2017*), the search terms included cooperative, collaborative, shared, team, collective, reasoning, problem solving, decision-making, combined with clinical or medicine or medical, but excluded shared decision-making (see Table S1). The primary interest of subjects was associated only with healthcare professionals who were involved actively in clinical activities. The studies involving patients or trainees such as students and interns were excluded.

3. Study selection and screening
   All papers were collected and managed using EndNote® software to eliminate duplicates. Initially, CYL and HYL screened only the title and abstract independently to filter articles that fail to meet the minimum inclusion criteria. All of the full-text articles were then reviewed by two additional researchers (CHL and MMC). The exclusion criteria were applied to non-peer-reviewed paper, conference, letters or editorial articles, papers lack of original data, and those without full-text available. Papers involved discussion mainly about individual clinical reasoning itself but without any types of team effort or collaborative interaction were also excluded.

4. Data charting
   Relevant papers were then imported to ATLAS.ti™ from EndNote® after screening. A charting content was developed using ATLAS.ti™ to ensure relevance and to extract study characteristics, including publication year, publication type, methodology, participant details (RQ 1). Additionally, critical findings germane to the exploration of multidisciplinary teams or meetings (MDTs or MDMs), encompassing composition and content aspects, were systematically extracted (RQ 2). This charting process was reviewed by the research team and pretested by all reviewers before implementation. The characteristics of each full-text article were extracted and coded by two independent reviewers (CYL and SYY). Studies failing to meet the eligibility criteria were further

excluded. Reviewers met throughout the process to resolve conflicts and ensured consistency with the research questions.

5. Data summary and synthesis

   To systematically analyze the collected data, a comprehensive approach blending quantitative and thematic methods was employed. This involved the development of an analytical framework to collate and interpret various themes derived from the gathered information. For the quantitative analysis, an overview of basic descriptive frequency counts was conducted, focusing on key article 'demographic,' such as publication year and journal. This quantitative lens facilitated a high-level understanding of the distribution and trends within the selected literature. Simultaneously, thematic coding was applied to extract and categorize the content of each article. This involved identifying recurring patterns, concepts, or topics relevant to collaborative clinical reasoning (CCR). Frequencies of counts were summarized and presented in graphical or tabulated form. Microsoft Excel 2010 (Microsoft, Redmond, WA, USA) was used to facilitate descriptive analyses and graphical summaries. Each article was coded by a maximum of two themes by ATLAS.ti™.

6. Team consultation

   The research members met on a weekly basis to track the progress of the scoping review, and monthly meetings were held with the international consultant for further consolidation of results.

## RESULTS

The initial database searches yielded 281 citations. After conducting a duplication check and screening titles and abstracts against the exclusion criteria, 24 articles met the eligibility criteria for comprehensive review and analysis (Fig. 1).

### Year, journal, and methodology (RQs 1–2)

The average frequency of the included articles on CCR ranged between 1 and 2 per year between 2011 and 2016 (Fig. 2). The ranged between 1 and 2 per year increased to 3 in 2017 but declined to 1 again in 2018. The highest and second highest number of CCR studies for analysis were found in 2019 ($n = 6$) and 2020 ($n = 4$), respectively. The journals with which these 24 articles were published were listed alphabetically in Table S2, found in SI. There were only 2 articles published in the same journal, Annals of Surgical Oncology. Each journal as suggested by its name was categorized into six genres. The majority of the articles fell into categories of oncology ($n = 8$) and medicine in general ($n = 7$) while the rest of the articles made up the categories of nursing ($n = 2$), medical education ($n = 3$), ergonomics or medical informatics ($n = 2$), and philosophy or psychology ($n = 2$). Both quantitative ($n = 11$) and qualitative ($n = 10$) methodology were the most prevalent approaches while mixed methods ($n = 3$) was the least common approach.

### Themes, population, and trends (RQs 3–5)

In Table 1, the matching of the articles into four major content themes were as follows: (1) Decision-making process ($n = 14$) (*Alby, Zucchermaglio & Baruzzo, 2015*; *Alcantara*

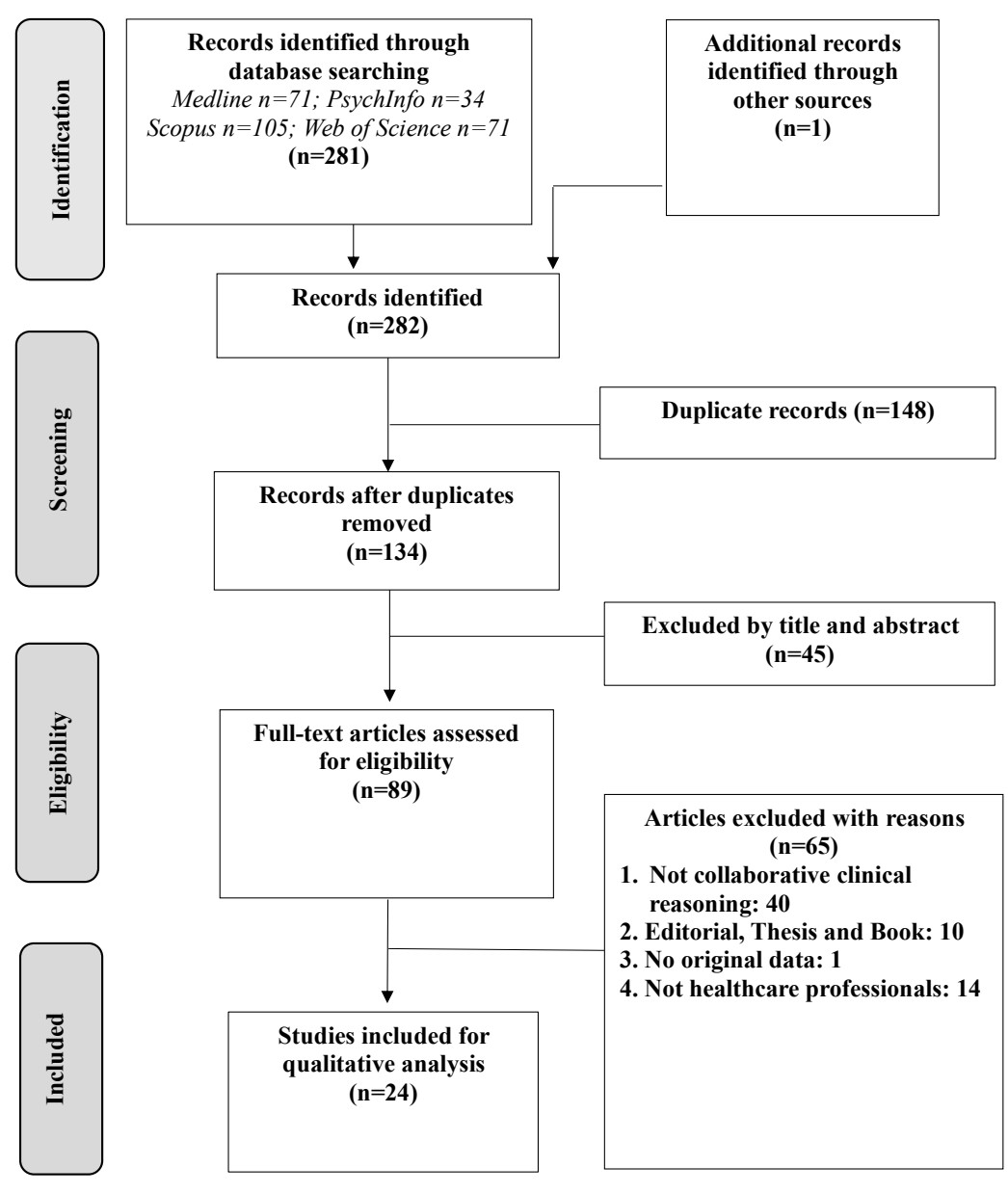

**Figure 1   Flowchart of the study selection process.**

*et al., 2014*; *Bingham et al., 2020*; *Bolle et al., 2019*; *Charani et al., 2019*; *Jalil et al., 2013*; *Kilpatrick, 2013*; *Kinnear, Wilson & O'Dwyer, 2018*; *Lamb et al., 2011*; *Lamb et al., 2012*; *Radcliffe et al., 2019*; *van Baalen & Carusi, 2019*; *Wallace et al., 2019*; *Wolf et al., 2015*); (2) quality assessment by MDTs-MODe (Multidisciplinary Team-Metric for the Observation of Decision Making; $n = 5$) (*Gandamihardja et al., 2019*; *Hahlweg et al., 2017*; *Scott et al., 2020*; *Soukup et al., 2020*; *Soukup et al., 2016*); (3) CCR theory and definitions ($n = 3$) (*Blondon et al., 2017*; *Kiesewetter, Fischer & Fischer, 2017*; *Olson et al., 2020*); and (4) problem-solving
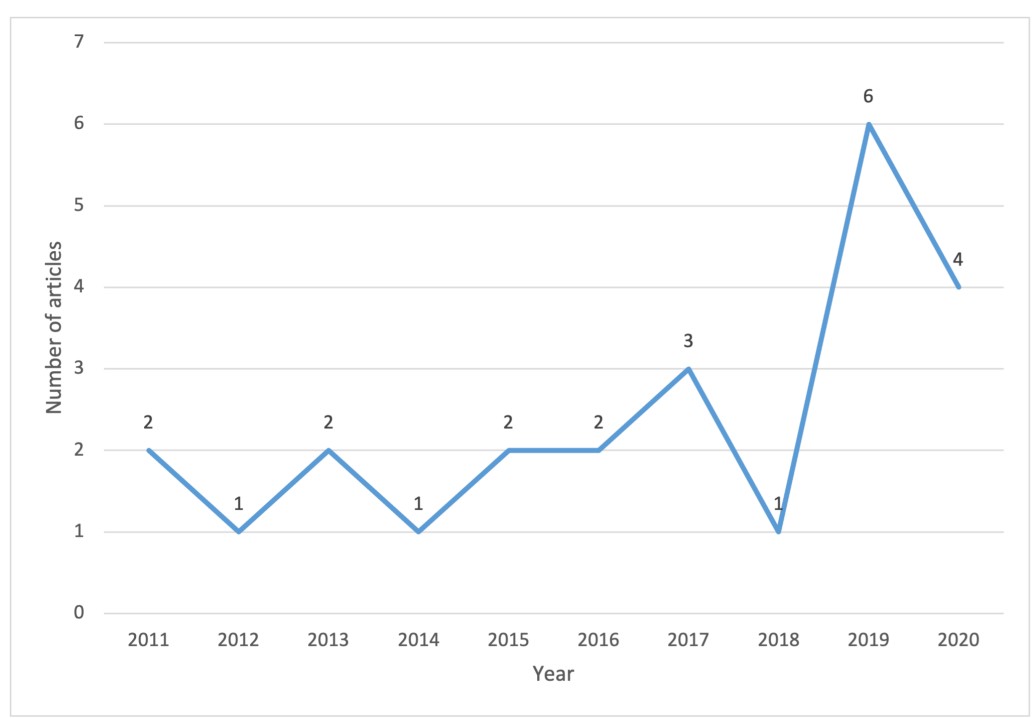

**Figure 2** The number of included articles on collaborative clinical reasoning between 2011 and 2020.

process ($n = 2$) (*Maseide, 2011*; *Måseide, 2016*). The double coding frequency was also used to support the key issues identified within content themes.

## MDT participants and data collection

Overall, there are 14 studies conducted with MDT members (*Alcantara et al., 2014*; *Bolle et al., 2019*; *Gandamihardja et al., 2019*; *Hahlweg et al., 2017*; *Jalil et al., 2013*; *Lamb et al., 2011*; *Lamb et al., 2012*; *Maseide, 2011*; *Måseide, 2016*; *Scott et al., 2020*; *Soukup et al., 2020*; *Soukup et al., 2016*; *van Baalen & Carusi, 2019*; *Wallace et al., 2019*). Only one article among these MDT-related studies collects both non-cancer and occasionally cancer related MDT data in a thoracic ward (*van Baalen & Carusi, 2019*). The remaining 13 articles all address issues about cancer MDT, five of which focus on MDT quality assessment utilising the tool, MDTs-MODe. The most discussed MDT case was colorectal or gastrointestinal cancer. In terms of the MDT composition, nurses or nurse specialists were the most frequently identified team members. The second and third highest proportion of team members, namely surgeons, radiologists, histopathologists and oncologists entails how they are often coupled with nurses or nurse specialists, and altogether they often represent the common composition of team members found in a cancer MDT.

## Non-MDT-specific articles

These studies do not specifically include the term MDT, however there are few of them do fall into the category of team concept. These studies are also summarized by minor themes (Table 1). Two reviews describe the theory about CCR (*Kiesewetter, Fischer &*

**Table 1   Content themes for articles on collaborative clinical reasoning between 2011–2020.**

| Major content themes | Single coding frequency |
|---|---|
| Decision-making process<br>Any article directly addresses the topic of "decision-making" process in the title or keyword, or as the subject of interest throughout the context (*Alby, Zuccermaglio & Baruzzo, 2015*; *Alcantara et al., 2014*; *Bingham et al., 2020*; *Bolle et al., 2019*; *Charani et al., 2019*; *Jalil et al., 2013*; *Kilpatrick, 2013*; *Kinnear, Wilson & O'Dwyer, 2018*; *Lamb et al., 2011*; *Lamb et al., 2012*; *Radcliffe et al., 2019*; *van Baalen & Carusi, 2019*; *Wallace et al., 2019*; *Wolf et al., 2015*). | 14 |
| Quality assessment by MDTs-MODe<br>Articles involve the assessment of multi-disciplinary team meetings (MDT or MDM) using the standard MDT-MODe (Multidisciplinary Team-Metric for the Observation of Decision Making) (*Gandamihardja et al., 2019*; *Hahlweg et al., 2017*; *Scott et al., 2020*; *Soukup et al., 2020*; *Soukup et al., 2016*). | 5 |
| Collaborative clinical reasoning theory and definitions<br>Articles specifically explain the theory or definitions about collaborative clinical reasoning (*Blondon et al., 2017*; *Kiesewetter, Fischer & Fischer, 2017*; *Olson et al., 2020*). | 3 |
| Problem-solving process<br>Articles directly address the topic of "problem solving" process in the title or keyword, or as the subject of interest throughout the context (*Maseide, 2011*; *Måseide, 2016*). | 2 |
| **Minor content themes** | **Double coding frequency** |
| Articles involve multi-disciplinary team meetings (MDT or MDM) (eg. 13 cancer MDTs, 1 thoracic) | 14 |
| Communication and other factors (eg. culture) in decision-making | 8 |
| Collective intelligence (eg. compositional team cognition) | 4 |
| Trigger for decision-making (eg. Nurses initiate decision-making) | 2 |
| Team conversational data (their relation to decision-making or problem solving) | 2 |
| Simulation in ward | 1 |

*Fischer, 2017*; *Olson et al., 2020*) while one review characterizes collective intelligence in medical decision-making (*Radcliffe et al., 2019*). Two comparative studies show evidence on better performance in teams than individuals when solving a cognitive drug problem (*Kinnear, Wilson & O'Dwyer, 2018*) or interpreting mammograph screening (*Wolf et al., 2015*). One study qualitatively compares the different decision-making process on antibody prescriptions between emergency and surgical teams, where the authors attribute such difference to team culture (*Charani et al., 2019*). One simulation study conducted with residents and nurses in internal medicine wards identifies characteristics and dimensions of CCR (*Blondon et al., 2017*). Two studies demonstrate the importance of communication during decision-making process, and specifically the role of a nursing staff

on initiating a decision-making process in a team (*Bingham et al., 2020*; *Kilpatrick, 2013*). Upon qualitative analysis of informal conversations about patient cases in a medical team, one study reveals three collaborative practices: (a) joint interpretation, (b) intersubjective generation and validation of hypotheses, and (c) postponing the diagnostic decision (*Alby, Zucchermaglio & Baruzzo, 2015*). In general, several articles have addressed separately how communication, trust, team composition, institutional culture, or prescriptive authority may exert an influence on collaborative practice in healthcare team decision-making (*Alby, Zucchermaglio & Baruzzo, 2015*; *Alcantara et al., 2014*; *Bingham et al., 2020*; *Blondon et al., 2017*; *Bolle et al., 2019*; *Charani et al., 2019*; *Jalil et al., 2013*; *Kilpatrick, 2013*; *Kinnear, Wilson & O'Dwyer, 2018*; *Måseide, 2016*; *van Baalen & Carusi, 2019*; *Wallace et al., 2019*).

# DISCUSSION

This scoping review illuminates the landscape of CCR research spanning 2011 to 2020, consisting of 24 identified studies. Notable trends in yearly publications reflect an initial alternation between one and two articles from 2011 to 2016, a peak in 2019, and a sustained level of interest in 2020. This temporal evolution underscores the growing importance and recognition of CCR research in recent years. The 24 selected articles spanned various journals, with only two articles appearing in the same journal (Annals of Surgical Oncology). These journals were categorized into six genres, predominantly falling within oncology and medicine in general. Methodologically, both quantitative and qualitative approaches were prevalent, with mixed methods being the least common. The remaining articles covered nursing, medical education, ergonomics or medical informatics, and philosophy or psychology.

A comprehensive analysis of these studies reveals distinct patterns and avenues for advancing understanding in this multidimensional field. Four major fields were identified including decision-making process, CCR theory and definitions, quality assessment by MDTs-MODe, and problem-solving process. The dominant theme was the decision-making process. The prevalence of studies emphasizing the decision-making process underscores its centrality in CCR. The articles focus on communication and factors associated with collaborative decision-making processes. However, the majority of discussion dwell on the conceptual importance of CCR, leaving a noticeable gap in the concrete definition, structure, and process characterizing CCR. In depth, only two studies provide explicit definitions and theoretical frameworks for CCR, elucidating key factors influencing its performance (*Blondon et al., 2017*; *Kiesewetter, Fischer & Fischer, 2017*). *Kiesewetter, Fischer & Fischer (2017)* summarized factors that may influence the performance of CCR: (1) The initial distribution of information, (2) practitioners' clinical experience in a team, (3) information exchange among members, and (4) individual retrieval and representation of the information that shared by a team such as distribution of information or clinical experience. In a simulation study conducted in healthcare setting, *Blondon et al. (2017)* have identified five dimensions of collaborative reasoning in internal medicine: (1) diagnostic reasoning, (2) patient management, (3) patient monitoring, (4) communication and (5) explanations to patient. Based on the

definitions of CCR from these two studies (*Blondon et al., 2017*; *Kiesewetter, Fischer & Fischer, 2017*), one review emphasizes the importance of clinical reasoning collaboration in relation to the development of shared decision-making or inter-professional education (*Hanum & Findyartini, 2020*). *Kiesewetter, Fischer & Fischer*'s (*2017*) focus on information distribution, clinical experience, and exchange, and *Blondon et al.*'s (*2017*) identification of five dimensions serve as foundational pillars, urging future research to integrate these frameworks for a deeper understanding. Integrating these conceptual frameworks into future research is essential for a more profound understanding of CCR.

Despite the prevalence of CCR studies, the broader landscape of interprofessional collaboration and clinical reasoning remains underexplored within the identified studies. The literature search in healthcare collaboration reveals terminologies such as interdisciplinary, multidisciplinary, interprofessional and intraprofessional, commonly interchangeably with teamwork, team approaches, collaborative practice, coordination and cooperation (*Angelini, 2011*; *D'Amour et al., 2005*; *Körner, 2010*; *Smith, 2015*). Only a handful of studies delve into the confluence of CCR and interprofessional collaboration, revealing nuances in role perceptions and expectations within healthcare teams (*Blondon et al., 2017*; *Hanum & Findyartini, 2020*; *Kiesewetter, Fischer & Fischer, 2017*; *Muller-Juge et al., 2013*; *Olson et al., 2020*; *Wölfel et al., 2016*). *Muller-Juge et al. (2013)* conducted semi-structured interviews with nurses and residents, exploring their role perceptions and expectations on interprofessional collaboration in an internal medicine ward. Their study highlighted a thematic findings wherein both professions perceived residents play a major role in clinical reasoning within collaborative framework (*Muller-Juge et al., 2013*). In a parallel context of internal medicine, nurses and physicians in another study by *Wölfel et al. (2016)* regarded CCR as core-competences and particularly essential for interprofessional development. However, in strict terms, these two studies exhibited limit relevance to CCR, despite acknowledging clinical reasoning as a fundamental component of collaborative practice.

While communication emerges as a critical dimension in CCR, akin to the findings of *Blondon et al. (2017)*, it is seldom explored comprehensively across the literature. *Olson et al. (2020)* observed that team clinical reasoning within existing healthcare often leads to a "parallel play" rather than authentic collaborative practice. Therefore, concerted efforts have been directed toward team communication, aiming to enhance information exchange and optimize decision-making during collaborative practice (*Lancaster et al., 2015*; *Matziou et al., 2012*). This observation is consistent with our scoping review, where a cluster of MDT studies showcases the utilization of MDT-MODe to assess information retrieval and communication among healthcare teams for evaluation of the decision-making quality. Although communication stands out as one of the dimensions identified in the CCR process (*Blondon et al., 2017*), other dimension, such as diagnostic reasoning, are infrequently explored across the literature.

Additional themes emerged, such as articles involving multi-disciplinary team meetings (MDT or MDM), communication and other factors in decision-making, collective intelligence, triggers for decision-making, team conversational data, and simulation in the ward. These minor themes provide a nuanced understanding of the factors influencing

CCR. Out of the 14 studies conducted with MDT members, the majority focused on cancer MDTs, particularly colorectal or gastrointestinal cancer. Nurses or nurse specialists were frequently identified team members, followed by surgeons, radiologists, histopathologists, and oncologists. This composition represented the common team structure in cancer MDTs. Studies not explicitly labeled as MDT-related fell into the broader category of team concepts.

## IMPLICATION

These articles covered diverse topics such as reviews on CCR research, comparative studies demonstrating team performance advantages, and simulation studies identifying characteristics of CCR. Communication, trust, team composition, institutional culture, and prescriptive authority were addressed as influencers in healthcare team decision-making. The scoping review highlights the versatility of CCR research, extending beyond healthcare into areas like digital public health interventions, dental care, and occupational therapy. This broad applicability emphasizes the comprehensive nature of CCR and its relevance across different disciplines. Future CCR research should aim for a more integrated understanding by incorporating explicit theoretical frameworks, such as those proposed by *Blondon et al. (2017)* and *Kiesewetter, Fischer & Fischer (2017)*. These frameworks will not only guide research design but also foster a nuanced interpretation of CCR dynamics within collaborative teams. This comprehensive approach will contribute to the evolution of evidence-based practices in collaborative clinical reasoning, fostering a more patient-centered and interprofessionally integrated healthcare landscape.

## CONCLUSIONS

This study provides the literature overview on CCR research spanning 2011 to 2020, revealing both a temporal evolution and a research diversity reflective of the multidimensional nature of CCR. The pronounced emphasis on the decision-making process within CCR, as evidenced by a prevalence of studies, underscores its central role. However, a discernible gap exists due to the absence of precise definitions and structures characterizing CCR. *Blondon et al. (2017)* and *Kiesewetter, Fischer & Fischer (2017)* provide explicit definitions and theoretical frameworks, serving as foundational pillars for future research integration. The call for future research to incorporate explicit theoretical frameworks, particularly those proposed by *Blondon et al. (2017)* and *Kiesewetter, Fischer & Fischer (2017)*, is crucial for guiding research design and interpreting CCR dynamics within collaborative teams. This integrated approach, including an awareness of the cognitive processes in CCR, aims to contribute to evidence-based practices in collaborative clinical reasoning, promoting a more patient-centered and interprofessionally integrated healthcare landscape.

### Funding

This work was supported by the National Science and Technology Council (R.O.C.) (grant number: NSTC 111-2628-H-182-001), and the Chang Gung Memorial Hospital in Taiwan (grant number: CMRPG3L1531). The funders had no role in study design, data collection and analysis, decision to publish, or preparation of the manuscript.

### Grant Disclosures

The following grant information was disclosed by the authors:
National Science and Technology Council (R.O.C.): NSTC 111-2628-H-182-001.
Chang Gung Memorial Hospital in Taiwan: CMRPG3L1531.

### Competing Interests

The authors declare there are no competing interests.

### Author Contributions

- Ching-Yi Lee conceived and designed the experiments, performed the experiments, analyzed the data, authored or reviewed drafts of the article, and approved the final draft.
- Hung-Yi Lai conceived and designed the experiments, performed the experiments, analyzed the data, authored or reviewed drafts of the article, and approved the final draft.
- Ching-Hsin Lee performed the experiments, analyzed the data, authored or reviewed drafts of the article, and approved the final draft.
- Mi-Mi Chen analyzed the data, prepared figures and/or tables, authored or reviewed drafts of the article, and approved the final draft.
- Sze-Yuen Yau analyzed the data, prepared figures and/or tables, authored or reviewed drafts of the article, and approved the final draft.

### Data Availability

This is a literature review.

### Supplemental Information

Supplemental information for this article can be found online at http://dx.doi.org/10.7717/peerj.17042#supplemental-information.

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
