# Peer review of "Collaborative clinical reasoning: a scoping review"

_PeerJ, doi:10.7717/peerj.17042_

## Round 0.1 · original submission · Major Revisions

Reviewers have considered your work and found some merits despite their great concerns. One recommended rejection, but as the editor, I strongly feel that the authors can elevate the quality of this work with the comments the reviewer provided. Please provide very strong details, not only in the revised manuscript but in the reply to reviewers' comments. Thank you

Reviewer 1 ·

Basic reporting

This is a highly relevant topic which merits a rigorous study. Overall the manuscript requires more work on a couple of areas to be able to contribute to the scientific community.
Background:
First sentence in “Clinical Reasoning in Medical Education” states that CR is a central component in the physician competence. The paragraph does not mention other professions which signals that the topic relates only to physicians. This is not stated in the title or aim, and further on the interprofessional CR is mentioned. So I lack some background on other professions relation to CR.
Later on in the text the interprofessional clinical reasoning is mentioned, so why the focus on only physicians here?
“Another theory of dual-processing argues…” I would argue that the dual-process theory acknowledges the two processes (system 1 AND 2), hence the “dual” in the name. Thus, the wording “another theory” is confusing.
Please introduce the MDT concept in the background, it is occurs quite abruptly in the result section.

Experimental design

The scoping review design is relevant but needs more clarification.

More details is needed on the screening process and inclusion of fulltexts: did one or several persons screen and decided on fulltext inclusion?
More details is needed on data extraction. What was extracted and how? (see also comment below on research questions)
Please also see and refer to PRISMA guidelines for reporting on scoping reviews: http://www.prisma-statement.org/Extensions/ScopingReviews
Better to place research questions directly after the aim in the previous section (background)
The additional RQ:s are better labelled as extraction aspects rather than a RQ.
Search strategy: from PRISMA: ”Present the full electronic search strategy for at least one database, including any limits used, such that it could be repeated”. This can be done as supplementary material if the journal provides such an option.

Data summary and synthesis:
“An analytical framework of quantitative and thematic approach was used… ”
This is very unspecific, please clarify/elaborate.

“Following data characterization, 24 articles were included for full coding (see Appendix 1 in the Supplementary Information, SI).”
The word “coding” is not the best option here because the data were not really coded?

Validity of the findings

Please include the PRISMA-chart in the result section, which is now provided in supplementary file. In doing so the first paragraph does not have to provide all details but refer to this figure.
Please revise wording (also in figure) concerning “annually published articles on CCR”. There are obviously more articles than these on the topic that are published, these numbers refer to the ones included for full-text analysis in your study.

Additional comments

Discussion/conclusion
Given the characteristics of Scoping review a more clear summary and discussion is needed on the state of the art in the field of CCR. Also recommendations for future developments in the research agenda for CCR. These aspects are also needed in the conclusion section.
The Donabedian model is briefly mentioned but not explained or elaborated. Please better consider how to use this model and if included it has to be introduced and used more clearly to discuss the results.
The section on Implication is confusing because it re-starts to introduce and teach about CR which is provided in the Background. Also very physician-oriented and not concerning the collaborative team including other professions (see comment above). This long paragraph has to be condensed and be focused towards implications if that is the label.

Cite this review as

Reviewer 2 ·

Basic reporting

I commend you for taking on the important work of performing a scoping review on a critical topic in the clinical reasoning literature: collaborative clinical reasoning. As a clinician-educator and education researcher interested in how the clinical reasoning process takes place within professional teams, I read this paper with great interest.

Experimental design

I appreciate your use of the Arksey and O’Malley framework for scoping reviews, which is a strength of this paper. My primary concerns relate to the manuscript’s ability to advance the literature on collaborative clinical reasoning due to its theoretical framing, presentation of results, and lack of alignment between the results and review questions.

You’ve effectively introduced an important cognitive theory related to clinical reasoning– dual-process theory. However, several other theories related to collaborative clinical reasoning are just as, if not more relevant, to the collaborative clinical reasoning process (e.g., situativity theories such as distributed cognition, ecological psychology, and situated cognition), some of which are mentioned in the articles you reviewed. While dual-process theory remains a key aspect of clinical reasoning performance, it focuses primarily on individual cognition. In addition, after the mention of dual-process theory in the introduction, it does not seem to play a primary role in your manuscript. Given the focus of this paper on collaborative and team-based clinical reasoning, your paper would be strengthened by a shift in the theoretical frameworks that underpin your approach to this field and consistent integration of them throughout your manuscript.

Validity of the findings

Your results currently organize articles primarily based on the population studied. Basing them instead on factors that align with the purposes of a scoping review (e.g., to summarize and disseminate research findings, identify gaps in the existing literature, clarify key concepts/definitions) would be a more effective format that can help the field of clinical reasoning scholarship understand the current state and potential future directions for collaborative clinical reasoning research. While your discussion does address some current conceptualizations of collaborative clinical reasoning and gaps in the current definitions, these ideas need to be further developed. Including this synthesis in your results, rather than the discussion, may allow for more detailed development of these ideas.

Finally, your results do not address your review questions in detail. For example, the portion of the results that addresses the review questions is primarily descriptive rather than synthetic. You mention publication dates, methodologies, and topics the articles cover, rather than summarize relevant theories or articulate descriptive themes that capture the current state of collaborative clinical reasoning research. Creating more alignment between your review questions and results would strengthen this paper by improving the cohesiveness between the background, methodology, and results.

Additional comments

You’ve clearly done a substantial amount of work to collect and review the current literature, and that work is not for nothing. With these changes, while they are substantial and will likely require much more analysis and literature review, this paper could be an important contribution to the field.

Cite this review as

---

## Round 0.2 · accepted · Accept

After a very thorough examination of the current revised version, I am very satisfied and convinced this work is acceptable for publication. The authors, I believe, diligently addressed the comments raised. Thank you for making PeerJ your journal of choice, and I look forward to your future scholarly contributions.
Congratulations

Reviewer 1 ·

Basic reporting

Dear editor,
Thank you for the possibility to review the revised version of the manuscript. The authors have done a good job in addressing my previous concerns and reactions. It is now more readable and enhanced in the scientific respect. I am happy to recommend publication of this version.

Experimental design

The design meets the expected standard.

Validity of the findings

The findings are valid

Additional comments

Good work, I consider this study to be a contribution to the scientific community on the topic of collaborative clinical reasoning.

Cite this review as